# Sources of Information about COVID-19 Vaccines for Children and Its Associations with Parental Motivation to Have Their Children Vaccinated in Taiwan

**DOI:** 10.3390/vaccines11081337

**Published:** 2023-08-07

**Authors:** Tai-Ling Liu, Ray C. Hsiao, Yu-Min Chen, Po-Chun Lin, Cheng-Fang Yen

**Affiliations:** 1Department of Psychiatry, Kaohsiung Medical University Hospital, Kaohsiung 80708, Taiwan; 2Department of Psychiatry, School of Medicine, Kaohsiung Medical University, Kaohsiung 80708, Taiwan; 3Department of Psychiatry and Behavioral Sciences, University of Washington School of Medicine, Seattle, WA 98195, USA; 4Department of Psychiatry, Seattle Children’s, Seattle, WA 98105, USA; 5Department of Psychiatry, E-Da Hospital, I-Shou University, Kaohsiung 82445, Taiwan; 6Department of Medicine, College of Medicine, I-Shou University, Kaohsiung 82445, Taiwan; 7College of Professional Studies, National Pingtung University of Science and Technology, Pingtung 91201, Taiwan

**Keywords:** child, COVID-19, information source, trust, vaccine

## Abstract

Pediatric COVID-19 vaccines have been developed to reduce the risk of contracting COVID-19 and subsequent hospitalization in children. Few studies have examined whether different sources of information regarding pediatric COVID-19 vaccines and parents’ trust in the information have different effects on parental motivation to have their child vaccinated. No study has examined parental demographic factors related to the sources of information and the trust of parents in these sources. Understanding the sources of information on pediatric COVID-19 vaccines, parents’ trust in the information, and related factors can contribute to the development of strategies for promoting the knowledge and acceptance of pediatric vaccination among parents. This study examined the sources of information regarding pediatric COVID-19 vaccines used by parents, their level of trust in these information sources, the demographic factors that influence this trust, and the associations of such information sources with parental motivation to get their child vaccinated against COVID-19. In total, 550 parents (123 men and 427 women) completed a questionnaire that was used to collect information regarding the information sources and to measure the parents’ trust in these information sources. Parental motivation to get their child vaccinated was measured using the Motors of COVID-19 Vaccination Acceptance Scale for Parents. Multivariate linear regression analysis was performed to examine two associations, namely the associations of the parents’ sources of information and their trust in these sources with their motivation to have their child vaccinated and the associations of the parents’ demographic factors with their sources of information and their trust in these sources. For the parents, traditional mass media and medical staff in healthcare settings were the most common sources of information regarding pediatric COVID-19 vaccines. The parents rated medical staff in healthcare settings as the most trustworthy source of information. Obtaining information from acquaintances through social media and obtaining information from medical staff in healthcare settings were significantly associated with parental motivation to get their child vaccinated against COVID-19. Trust in the information provided by medical staff in healthcare settings and coworkers was significantly associated with the motivation of parents to vaccinate their children against COVID-19. Compared with fathers, mothers were more likely to obtain information from medical staff in healthcare settings and from acquaintances through social media. Parents with a higher education level were more likely to obtain information from medical staff in healthcare settings. Compared with the fathers, the mothers were more trusting of information obtained from coworkers. Health professionals should consider the sources of information used by parents and related factors when establishing strategies to increase parental motivation to get their children vaccinated against COVID-19.

## 1. Introduction

The coronavirus disease 2019 (COVID-19) pandemic has had a serious impact on human health and daily life. According to the World Health Organization (WHO), by 14 July 2023, more than 760 million people had been confirmed to contract COVID-19; nearly 7 million people have died from COVID-19 infection [1]. Survivors of COVID-19 are at risk for a variety of post-acute sequelae of COVID-19. A meta-analysis of 41 published studies demonstrated that the global estimated pooled prevalence of post-acute sequelae of COVID-19 was 43% [2]; fatigue was the most common symptom reported, followed by memory problems [3]. In addition, the COVID-19 pandemic has negatively impacted the accessibility and structure of education, availability, and prices of food, economic activities, mental health, and quality of life in people around the world [4,5,6]. Studies have reported that children have a lower risk of developing a severe illness due to COVID-19 than adults [7,8]. However, preventing COVID-19 infection in children is crucial for several reasons. First, children with COVID-19 are at risk of developing multisystem inflammatory syndrome, which is a rare but serious complication of COVID-19 [9]. Second, 1–25% of children with COVID-19 develop long COVID, which may have prolonged effects on their health and daily lives [10]. Third, children play an important role in the transmission of COVID-19 in various settings (e.g., schools and communities) [11]. Although the World Health Organization (WHO) declared on 3 May 2023, that COVID-19 is no longer an international public health emergency [12], an additional 6848 child COVID-19 cases were reported on 11 May 2023 [13]. Children older than 6 months can receive vaccination against COVID-19 now [14]. Vaccinated children have a lower risk of being infected by COVID-19 and subsequent hospitalization [14,15,16]. Therefore, assessing parental motivation to get their child vaccinated against COVID-19 and the factors that influence this decision and conducting intervention to increase the rate of vaccine uptake against COVID-19 in children are necessary.

Most parents are motivated to have their children vaccinated to reduce the possibility of being infected by COVID-19; however, numerous parents are hesitant about doing so [17,18,19,20,21,22,23]. A meta-analysis of 44 published studies on parental attitudes toward pediatric COVID-19 vaccines revealed that three-fifths of parents were willing to have their children vaccinated, whereas 22.9% refused pediatric vaccination against COVID-19, and one-fourth were unsure [24]. Various factors influencing parental attitudes toward pediatric vaccination have been identified; they include parental sex, age, income, education level, the perceived threat from COVID-19, attitudes toward vaccination, the effects and side effects of vaccines, and public attitudes [24,25,26].

Studies have applied protection motivation theory (PMT) [27,28] to examine the factors that influence individuals’ motivation to undergo COVID-19 vaccination [29,30]. According to PMT, higher perceived threats from COVID-19, high efficiency in preventing the spread of COVID-19, and high self-efficacy in receiving vaccines were significantly associated with less skepticism toward vaccine mandates [29,30]. Believing in the safety and response efficacy of vaccines is positively related to the intention to be vaccinated against COVID-19 [30]. PMT-based studies have also discovered that sources of information play a key role in individuals’ attitudes and behaviors pertaining to vaccination [31,32,33,34]. Parents tended to obtain information on pediatric COVID-19 vaccines from multiple sources. A study reported that for Chinese parents, new media established by the government, mass media, and key opinion leaders were the main sources of information on pediatric COVID-19 vaccines [35]. Another study discovered that among Polish parents, the media (e.g., the Internet and television) were the main sources of information on pediatric COVID-19 vaccines [36]. However, misinformation regarding COVID-19 vaccines is prevalent [37], especially on social media [38,39,40]. The spread of misinformation on COVID-19 vaccines, such as that pertaining to the development and safety of these vaccines, on social media results in hesitancy [38,40].

Several topics related to the sources of information regarding pediatric COVID-19 vaccines used by parents warrant further study. First, few studies have examined whether different sources of information regarding pediatric COVID-19 vaccines have different effects on the motivation of parents to vaccinate their children. Second, parents may use but not necessarily trust a source of information on COVID-19 vaccines. A study reported that the trust of parents in the information provided by the government, healthcare providers, and social media was significantly associated with their acceptance of pediatric COVID-19 vaccines [41]. Whether the subjective trust of parents in various sources of information influences their motivation to vaccinate their children requires further clarification. Third, no study has examined parental demographic factors related to the sources of information on pediatric COVID-19 vaccines and the trust of parents in these sources. Thus, identifying and understanding the parental factors that affect sources of information can contribute to the development of strategies for promoting the knowledge and acceptance of pediatric vaccination among parents. Fourth, parents’ motivation to have their child receive COVID-19 vaccine injection has not been examined using a validated scale. The Motors of COVID-19 Vaccination Acceptance Scale for Parents (P-MoVac-COVID19S) [42] is a new scale developed for measuring parents’ motivation to get their child vaccinated against COVID-19 and can be used for assessing parental motivation to let their child receive vaccine injection.

The present study had three objectives. The first objective was to identify the sources from which parents in Taiwan obtain information regarding pediatric COVID-19 vaccines and their level of trust in these sources. The second was to examine the associations of parent-related demographical factors with the sources of information used by parents and their trust in these sources. Third, this study examined the associations between these sources of information and the trust of parents in these sources with their motivation to vaccinate their children against COVID-19. We hypothesized that parents would obtain information on pediatric COVID-19 vaccines from multiple sources and exhibit various levels of trust in the sources. We hypothesized that the sex, age, and education level of parents would be significantly associated with the sources of information and level of trust in these sources. Third, we hypothesized that specific sources of information (e.g., medical professionals) used by parents and their level of trust in the sources would be significantly associated with stronger parental motivation to get their child vaccinated against COVID-19.

## 2. Methods

### 2.1. Participants and Procedure

The present study applied an online survey study design. We recruited the participants by posting advertisements on several social media platforms (e.g., Twitter, Facebook, and LINE) between August 2022 and April 2023. This study included Taiwanese parents who were older than or equal to 20 years of age and had children between the ages of 6 and 18. Those with impaired intellect, schizophrenia, cognitive problems due to head injury, major physical problems, or substance use problems were excluded from the present study. In total, 562 parents contacted the research assistants and were interested in this study. Twelve parents with children whose age was less than 6 years old or more than 18 years old were excluded. Thus, 550 parents participated in this study and provided informed consent before the assessment. The Institutional Review Board of Kaohsiung Medical University Hospital approved this study (approval number: KMUHIRB-KMUHIRB-E(I)-20220107).

### 2.2. Measures

#### 2.2.1. Sources of Information on Pediatric COVID-19 Vaccines and Trust in Information

The participants were asked whether they often obtained information on pediatric COVID-19 vaccines from seven sources, namely medical staff in healthcare settings, traditional mass media (e.g., broadcast media and newspapers), family members, close friends, coworkers, acquaintances on social media, and strangers on social media. A participant could respond to each item with a “yes” or “no” answer. Each participant was also asked how much they trusted the information that they obtained from each source. Each item was rated on a 5-point Likert scale with endpoints ranging from 1 (very low) to 5 (very high).

#### 2.2.2. P-MoVac-COVID19S

The nine-item P-MoVac-COVID19S [42] is built on a robust theoretical foundation, being an adaption of the well-established Motors of COVID-19 Vaccination Acceptance Scale [43,44], which is based on the cognitive model of empowerment (CME) [45]. In the CME, empowerment is considered an innate motivation that drives people to engage in purposeful behavior. This motivation originates from four distinct cognitive processes, namely value formation (e.g., considering the purposes of vaccinating children and the value of this behavior), impact comprehension (e.g., the belief that COVID-19 vaccination is effective in protecting children against COVID-19 infection and reducing the risk of severe symptoms), knowledge acquisition (e.g., comprehension of information and knowledge of pediatric COVID-19 vaccination), and a sense of autonomy (e.g., the feeling of being able to freely decide whether to get their child vaccinated) [42]. The P-MoVac-COVID19S is rated on a 7-point Likert-type scale with endpoints ranging from 1 (strongly disagree) to 7 (strongly agree). A higher total score indicates a higher parental motivation to have their children receive vaccination against COVID-19. The nine-item P-MoVac-COVID19S has a single-factor structure, favorable reliability, and validity [42]. The Cronbach’s α of the P-MoVac-COVID19S in this study was 0.897.

#### 2.2.3. Demographic Characteristics

Data on the participants’ sex, age, number of years of education, the sex and age of their children, and the total number of children aged between 6 to 18 in the household were collected.

### 2.3. Data Analysis

SPSS 22.0 (SPSS, Chicago, IL, USA) was used for data analysis. Descriptive statistics were used to summarize the characteristics of the participants, sources of information used for obtaining pediatric vaccines against COVID-19 and trust in the information, and their level of motivation to vaccinate their child. Multiple regression models were employed, and the demographics of the participants and their children were controlled to examine how the participants’ sources of information and their trust in the information obtained from these sources were associated with their motivation to vaccinate their children. Because significant collinearity was identified, stepwise multiple regression was performed. Multivariate linear regression was performed to examine the influence of the participants’ sex, age, educational level, and number of children in the household on the associations of their sources of information and their trust in such information with their motivation to vaccinate their children. A two-tailed *p*-value of <0.05 indicated statistical significance.

## 3. Results

The data pertaining to the participants’ demographics and their level of motivation to vaccinate their children are presented in Table 1. The average age of the participants and their children was 44.3 (standard deviation [SD] = 5.2 years) and 11.8 years (SD = 3.6 years), respectively. The majority of the participants were women (*n* = 427; 77.6%), and the participants’ children comprised slightly more boys than girls. The mean number of children in the household was 1.7 (SD = 0.6). The participants had an average of 16.3 years of education (SD = 2.5 years), and their mean level of motivation to get their child vaccinated against COVID-19 was 49.5 (SD = 7.9).

Table 2 presents the sources from which participants obtained information regarding pediatric COVID-19 vaccines and the participants’ level of trust in the information from various sources. The results indicate that traditional mass media and medical staff in healthcare settings were the most common sources of information regarding pediatric COVID-19 vaccines, followed by coworkers, acquaintances on social media, family members, close friends, and strangers on social media. The participants rated medical staff in healthcare settings as the most trustworthy source of information, followed by traditional mass media, family members, coworkers, close friends, acquaintances on social media, and strangers on social media.

Table 3 presents the results of the stepwise multiple regression analysis conducted to examine the associations of the participants’ sources of information and their trust in these sources with their motivation to have their children receive COVID-19 vaccine injection. The results of Model I indicate that obtaining information from acquaintances on social media and obtaining information from medical staff in healthcare settings were significantly associated with parental motivation. The results of Model II indicate that greater trust in the information obtained from medical staff in healthcare settings and coworkers was significantly associated with a greater parental motivation to vaccinate.

Table 4 presents the results pertaining to the associations of the participants’ sex, age, and education level with the acquisition of information from acquaintances on social media and medical staff in healthcare settings. Table 5 presents the results pertaining to the associations of the participants’ sex, age, and education level with their trust in the information obtained from medical staff in healthcare settings and coworkers. The results indicate that compared with the male participants, the female participants were more likely to obtain information from medical staff in healthcare settings and acquaintances on social media. The participants with a higher education level were more likely to obtain information from medical staff in healthcare settings. Compared with the male participants, the female participants were more trusting of the information obtained from coworkers. Participants’ ages and the number of children in the household were not significantly associated with the sources of information and participants’ trust.

## 4. Discussion

The present study revealed that parents obtain information on pediatric COVID-19 vaccines from multiple sources. This result was congruent with the result of a previous in Chinese parents [33]. This study further extended the scope of the study to examine the various effects of information obtained from different sources on parental motivation to get their child to receive vaccine injections. Although traditional mass media were the most common sources of information regarding pediatric COVID-19 vaccines, the participants rated medical staff in healthcare settings as the most trustworthy source of information. Obtaining information from acquaintances on social media and medical staff in health-care settings and the level of trust in the information obtained from medical staff in health-care settings and coworkers were significantly associated with the participants’ motivation to get their child vaccinated against COVID-19. Sex differences were identified in the participants’ sources of information and their level of trust in these sources.

Mass media, such as television and radio, are key channels by which the Taiwanese government promotes the importance of vaccines to the public and encourages vaccination to reduce the spread of COVID-19. For example, the Taiwanese government collaborated with medical experts to produce videos that were extensively aired on television. Accordingly, the participants in the present study most often obtained information regarding pediatric COVID-19 vaccines from traditional mass media. However, the participants were most trusting of the information on pediatric COVID-19 vaccines obtained from medical staff in healthcare settings. The participants’ acquisition of information from medical staff in healthcare settings and their trust in such information were significantly associated with their motivation to get their children vaccinated. The development of pediatric COVID-19 vaccines occurred over a short period. By contrast with other pediatric vaccines that require a long time to develop, pediatric COVID-19 vaccines are less understood by parents, who may not have adequate experience and knowledge to evaluate the advantages and disadvantages of such vaccines. The participants were more trusting of medical professionals whom they knew and interacted with before the pandemic than of medical professionals who disseminated knowledge regarding pediatric COVID-19 vaccines through traditional mass media; accordingly, the participants were more trusting of the information obtained from the medical professional whom they knew. The results indicate that in addition to inviting medical experts to disseminate knowledge and emphasize the importance of pediatric COVID-19 vaccines through mass media, the government should train medical staff in healthcare settings to strengthen their ability to answer parents’ questions and to increase the motivation of parents to vaccinate their child against COVID-19. The present study demonstrated that a higher education level was significantly associated with the acquisition of information on pediatric COVID-19 vaccines from medical staff in healthcare settings. Accordingly, medical staff must proactively provide information regarding vaccines and explain them in a manner that all parents can understand, regardless of their education level.

A previous study on Polish parents categorized television and the Internet as the same source of information [36], whereas the present study found that the influences of information obtained from traditional and social media on parental motivation to get their child vaccinated varied; moreover, the influences of information obtained from acquaintances and strangers on social media on parents’ motivation were also different. The present study revealed that obtaining information on pediatric COVID-19 vaccines from acquaintances on social media was significantly associated with the motivation of parents to vaccinate their children; however, this significant association was not identified in the association between the acquisition of information from strangers on social media and the motivation of parents. Although the participants were less trusting of the information obtained from acquaintances on social media than of that obtained from medical staff in healthcare settings, the acquisition of information from acquaintances on social media still influenced the participants’ motivation. During the pandemic, social media was a key source of information on COVID-19 and its vaccines. However, misinformation pertaining to COVID-19 vaccines is prevalent in social media, and it can increase the risk of vaccine hesitancy [38,39,40]. Thus, helping parents avoid being deceived by false messages from acquaintances on social media is crucial. The present study also discovered that the participants’ trust in the information obtained from coworkers was significantly associated with their motivation to get their children vaccinated. This result further supports the hypothesis that the trust of parents in vaccine-related information is influenced by their familiarity with their source of the information. Disseminating accurate information regarding COVID-19 vaccines to the general public and reducing their suspicion regarding vaccines can help to enhance the motivation of parents to vaccinate their children.

The present study discovered that the female participants were more likely than the male participants to obtain information on pediatric COVID-19 vaccines from medical staff in healthcare settings and acquaintances in social media; the female participants were also more trusting than the male participants of the information obtained from coworkers. A significantly greater proportion of the participants in the present study were women. Mothers are still the predominant caregivers of children in Taiwan. Compared with fathers, mothers may have more opportunities and exhibit a greater intention to engage in discussions with medical staff in healthcare settings and with their friends on social media, and this could have led to the sex imbalance of the participants in the present study. Whether parents should vaccinate their children against COVID-19 is a medical decision that they must consider; therefore, fathers should be encouraged to discuss vaccinating their children against COVID-19 with health-care providers.

Several limitations of the present study must be highlighted. First, because the present study collected data from a single source (parents), its results could have been affected by shared-method variance and social desirability bias. In addition, the participants were parents who were interested in the aim of this study and who may have been concerned about various aspects of pediatric COVID-19 vaccines. Second, the participants were recruited using online advertisements; therefore, they might not be representative of all parents in Taiwan. For example, the proportion of female participants in the study sample was significantly higher. Third, several factors that were not evaluated in the present study (e.g., the severity of the COVID-19 outbreak and reports on vaccine side effects) could have influenced the association between the sources of information of parents and their motivation to have their children vaccinated.

## 5. Conclusions

The present study demonstrated that parents obtain information on pediatric COVID-19 vaccines from multiple sources and exhibit different levels of trust in information from different sources. The information sources of the surveyed parents and their trust in the information from these sources were associated with their motivation to vaccinate their children. The sex and education level of the parents were significantly associated with the acquisition of information from specific sources and their trust in such information. Thus, healthcare providers should develop programs to improve parental motivation to get their child vaccinated by accessing the sources of information identified in the present study.

## Figures and Tables

**Table 1 vaccines-11-01337-t001:** Participants’ characteristics (N = 550).

	*n* (%)	M (SD)	Range
Parent sex			
Male	123 (22.4)		
Female	427 (77.6)		
Parent age (year)		44.3 (5.2)	28–67
Parent educational year		16.3 (2.5)	9–25
Child sex			
Male	301 (54.7)		
Female	249 (45.3)		
Child age (year)		11.8 (3.6)	6–18
Number of children in the household		1.7 (0.6)	1–4
Parental motivation to have their child vaccinated		49.5 (7.9)	12–63

**Table 2 vaccines-11-01337-t002:** Sources of information for pediatric COVID-19 vaccines and the trust participants had.

	Commonly Used	Trust	
	*n* (%)	M (SD)	Range
Traditional mass media	403 (73.3)	3.7 (1.0)	1–5
Medical staff in healthcare settings	312 (56.7)	4.4 (0.8)	1–5
Coworkers	245 (44.5)	3.4 (0.9)	1–5
Acquaintances on social media	204 (37.1)	2.7 (1.0)	1–5
Family members	193 (35.1)	3.5 (0.9)	1–5
Close friends	173 (31.5)	3.3 (0.9)	1–5
Strangers on social media	115 (20.9)	2.2 (1.0)	1–5

**Table 3 vaccines-11-01337-t003:** Associations of sources of information for pediatric COVID-19 vaccines and their trust with parental motivation to have their child vaccinated: stepwise multiple regression analysis.

	Parental Motivation			
	Model I		Model II	
	B (SE)	*p*	B (SE)	*p*
Source: Acquaintances on social media	3.014 (0.695)	<0.001		
Source: Medical staff in healthcare settings	1.466 (0.678)	0.031		
Trust: Medical staff in healthcare settings			2.642 (0.422)	<0.001
Trust: Coworkers			1.078 (0.377)	0.004
F	13.819		34.057	
*p*	<0.001		<0.001	
Adjusted R^2^	0.045		0.107	

**Table 4 vaccines-11-01337-t004:** Associations of parental sex, age, and education with sources of information for pediatric COVID-19 vaccines: logistic regression analysis.

	Medical Staff in Healthcare Settings		Acquaintances on Social Media	
	OR (95% CI)	*p*	OR (95% CI)	*p*
Sex ^a^	0.441 (0.288–0.675)	<0.001	0.590 (0.376–0.924)	0.021
Age	0.972 (0.939–1.006)	0.104	1.013 (0.978–1.049)	0.473
Education	1.148 (1.066–1.236)	<0.001	1.064 (0.990–1.143)	0.091
Number of children in the household	1.170 (0.874–1.568)	0.292	1.309 (0.977–1.754)	0.071

CI: confidence interval; OR: odds ratio. ^a^: Reference: female.

**Table 5 vaccines-11-01337-t005:** Associations of parental sex, age, and education with trust in information sources for pediatric COVID-19 vaccines: multiple regression analysis.

	Medical Staff in Healthcare Settings		Coworkers	
	B (SE)	*p*	B (SE)	*p*
Sex ^a^	–0.032 (0.084)	0.700	–0.299 (0.093)	0.001
Age	0.002 (0.007)	0.804	0.005 (0.008)	0.543
Education	0.021 (0.014)	0.141	0.031 (0.016)	0.054
Number of children in the household	0.021 (0.058)	0.719	0.075 (0.064)	0.238

CI: confidence interval; OR: odds ratio. ^a^: Reference: female.

## Data Availability

The data will be available upon reasonable request to the corresponding authors.

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
