# Peer review of "Sources of Information about COVID-19 Vaccines for Children and Its Associations with Parental Motivation to Have Their Children Vaccinated in Taiwan"

_vaccines, 2023, doi:10.3390/vaccines11081337_

Round 1

Reviewer 1 Report

This is a good report of a large survey of parents in Taiwan, regarding their likelihood to approve Covid vaccine for their children, related to their sources of information and their trust in those sources.   You have provided a thorough report of your methods and findings, and this supports other data in this field, and also provides ideas for interventions, to address parental concerns.  You address your limitations, specifically noting correctly that this is a convenience sample (volunteers) and since mostly women, differences between men and women may not be relevant. 

In Table 3, you present findings from two models.  I could not find in methods a description of the models.  It looks like in one, the outcome was source of info, and in the other, trust in the source.  Did you control for source in the model about trust?

Specific comments:

In the abstract be careful to note when you mean male or female parent (as opposed to child).  I was confused twice (line 7 from the top and line 4 from the end of the abstract). 

How did you handle parents with more than one child in the age group? Did you collect that information, and if so, did it make a difference?

Be careful in the Introduction to not imply that Covid vaccine prevents the infection.  It reduces risks related to the infection. 

Some edits will be needed to make sure English is clear. E.g. in conclusion: "...some sources and the trust..." should be "some sources and the trust in those sources... "

In abstract: "to prevent them from contracting, or being infected with SARS CoV 2.. or becoming ill with Covid 19.  

There are many small edits that can be made. 

Author Response

We appreciated your valuable comment. As discussed below, we have revised our manuscript with underlines based on your suggestions. We also invited an English native editor to edit the manuscript. Attached please find the certificate for the edition. Please let us know if we need to provide anything else regarding this revision.

Comment 1

This is a good report of a large survey of parents in Taiwan, regarding their likelihood to approve Covid vaccine for their children, related to their sources of information and their trust in those sources.   You have provided a thorough report of your methods and findings, and this supports other data in this field, and also provides ideas for interventions, to address parental concerns.  You address your limitations, specifically noting correctly that this is a convenience sample (volunteers) and since mostly women, differences between men and women may not be relevant. 

Response

Thank you for your comment. Indeed, the difference in the ratio of female and male participants in this study must be noted. We encouraged the father of children to raise awareness and discussions about pediatric COVID-19 vaccines as below. Please refer to line 318-320.

Whether parents should vaccinate their children against COVID-19 is a medical decision that they must consider; therefore, fathers should be encouraged to discuss about vaccinating their children against COVID-19 with health-care providers.

Comment 2

In Table 3, you present findings from two models.  I could not find in methods a description of the models.  It looks like in one, the outcome was source of info, and in the other, trust in the source.  Did you control for source in the model about trust?

Response

We revised the description regarding the statistical analysis as below. Please refer to line 184-188.

Multiple regression models were employed, and the demographics of the participants and their children were controlled to examine how the participants’ sources of information and their trust in the information obtained from these sources were associated with their motivation to vaccinate their children. Because significant collinearity was identified, stepwise multiple regression was performed.

Comment 3

In the abstract be careful to note when you mean male or female parent (as opposed to child).  I was confused twice (line 7 from the top and line 4 from the end of the abstract). 

Response

Thank you for your reminding. We revised the descriptions in Abstract as below.

“…550 parents (123 men and 427 women)…” Please refer to line 25-26.

“Compared with fathers, mothers were more likely to…” Please refer to line 40.

Comment 4

How did you handle parents with more than one child in the age group? Did you collect that information, and if so, did it make a difference?

Response

Yes, we have collected the total number of children aged between 6 to 18 in the household. In the revised manuscript we added the data as below. We also examined its associations with parental motivation to vaccinate their child; the results of stepwise multiple regression analysis indicated that the total number of children was not significantly with parental motivation (Table 3). We also examined its associations with the sources of information for pediatric COVID-19 vaccines and the trust participants had; the results of multiple logistic regression and multiple regression analysis indicated that the total number of children was not significantly with the source of information and participants’ trust (Tables 4 and 5).

  • “…the total number of children aged between 6 to 18 in the household…” Please refer to line 178.
  • The mean number of children in the household was 1.7 (SD = 0.6).” Please refer to line 198-199.
  • “Multivariate linear regression was performed to examine the influence of the participants sex, age, educational level, and number of children in the household on the associations…” Please refer to line XXX.
  • Participants’ age and the number of children in the household were not significantly associated with the sources of information and participants’ trust.” Please refer to line 190.
  •  

Comment 5

Be careful in the Introduction to not imply that Covid vaccine prevents the infection.  It reduces risks related to the infection. 

Response

Thank you for your comment. We revised this sentence as below. Please refer to line 70-71.

Vaccinating children against COVID-19 is a strategy for reducing the risk of contracting COVID-19 in this population [14].

Comment 6

Comments on the Quality of English Language: There are many small edits that can be made.

Response

We invited an English native editor to edit the manuscript. The grammar errors and typos were corrected. Attached please find the certificate for the edition.

Comment 6-1

Some edits will be needed to make sure English is clear. E.g. in conclusion: "...some sources and the trust..." should be "some sources and the trust in those sources... "

Response

We revised this sentence into “The sex and education level of the parents were significantly associated with the acquisition of information from specific sources and their trust in such information.” Please refer to line 337-339.

Comment 6-2

In abstract: "to prevent them from contracting, or being infected with SARS CoV 2. or becoming ill with Covid 19. 

Response

We revised this sentence into “Pediatric COVID-19 vaccines have been developed to prevent COVID-19 infection in children.” Please refer to line 15.

Reviewer 2 Report

1.The abstract can be improved by including the existing challenges, motivations and outcomes of the paper.
2. In the introduction section. the objective of the study needs to be clearly stated. Kindly strengthen the significance of the study. It is weak in its present form.
3. Kindly incorporate the research gap in the literature review. Some literature should be considered and referred.

4.sections of Introduction. The background of the COVID-19 pandemic seems too simple. The background of the impact of the COVID-19 pandemic needs to be considered. There is a need to better elaborate the background of the COVID-19 pandemic. Please consider citing following papers: entitled "A preliminary assessment of the impact of COVID-19 on environment–A case study of China", and entitled "What does the China's economic recovery after COVID-19 pandemic mean for the economic growth and energy consumption of other countries?".

5.section of Method. This section offers a detailed description. It is impressive. However, it would be better to highlight your improvement of the method and your innovation in methods.

6. section of Result. This section is well-organized and well-written. It would be better to discuss what your findings are different from the past works. Deep and systematic analysis about the results should highlight the novelty of this manuscript.

7.There are still some occasional grammar errors through the manuscript especially the article ''the'', ''a'' and ''an'' is missing in many places, please make a spellchecking in addition to these minor issues.

1.The abstract can be improved by including the existing challenges, motivations and outcomes of the paper.
2. In the introduction section. the objective of the study needs to be clearly stated. Kindly strengthen the significance of the study. It is weak in its present form.
3. Kindly incorporate the research gap in the literature review. Some literature should be considered and referred.

4.sections of Introduction. The background of the COVID-19 pandemic seems too simple. The background of the impact of the COVID-19 pandemic needs to be considered. There is a need to better elaborate the background of the COVID-19 pandemic. Please consider citing following papers: entitled "A preliminary assessment of the impact of COVID-19 on environment–A case study of China", and entitled "What does the China's economic recovery after COVID-19 pandemic mean for the economic growth and energy consumption of other countries?".

5.section of Method. This section offers a detailed description. It is impressive. However, it would be better to highlight your improvement of the method and your innovation in methods.

6. section of Result. This section is well-organized and well-written. It would be better to discuss what your findings are different from the past works. Deep and systematic analysis about the results should highlight the novelty of this manuscript.

7.There are still some occasional grammar errors through the manuscript especially the article ''the'', ''a'' and ''an'' is missing in many places, please make a spellchecking in addition to these minor issues.

Author Response

We appreciated your valuable comment. As discussed below, we have revised our manuscript with underlines based on your suggestions. We also invited an English native editor to edit the manuscript. Attached please find the certificate for the edition. Please let us know if we need to provide anything else regarding this revision.

Comment

1.The abstract can be improved by including the existing challenges, motivations and outcomes of the paper.
Response

Thank you for your suggestion. We revised the abstract as below to including the existing challenges, motivations and outcomes of the paper. Please refer to line 15-22.

Pediatric COVID-19 vaccines have been developed to prevent COVID-19 infection in children. Few studies have examined whether different sources of information regarding pediatric COVID-19 vaccines and parents’ trust in the information have different effects on the motivation of parents to vaccinate their children. No study has examined parental demographic factors related to the sources of information and the trust of parents in these sources. Understanding the sources of information on pediatric COVID-19 vaccines, parents’ trust in the information, and related factors can contribute to the development of strategies for promoting the knowledge and acceptance of pediatric vaccination among parents.

Comment

  1. In the introduction section. the objective of the study needs to be clearly stated. Kindly strengthen the significance of the study. It is weak in its present form.
    Response

Thank you for your comment. We described the objective of the study as below. Please refer to line 123-129.

The present study had three objectives. The first objective was to identify the sources from which parents in Taiwan obtain information regarding pediatric COVID-19 vaccines and their level of trust in these sources. The second was to examine the associations of parent-related demographical factors with the sources of information used by parents and their trust in these sources. Third, this study examined the associations of these sources of information and the trust of parents in these sources with their motivation to vaccinate their children against COVID-19.

Comment

  1. Kindly incorporate the research gap in the literature review. Some literature should be considered and referred.

Response

We described the research gap in the literature review as below. Please refer to line 103-122.

Several topics related to the sources of information regarding pediatric COVID-19 vaccines used by parents warrant further study. First, few studies have examined whether different sources of information regarding pediatric COVID-19 vaccines have different effects on the motivation of parents to vaccinate their children. Second, parents may use but not necessarily trust a source of information on COVID-19 vaccines. A study reported that the trust of parents in the information provided by the government, health-care providers and social media was significantly associated with their acceptance of pediatric COVID-19 vaccines [45]. Whether the subjective trust of parents in various sources of information influences their motivation to vaccinate their children requires further clarification. Third, no study has examined parental demographic factors related to the sources of information on pediatric COVID-19 vaccines and the trust of parents in these sources. Thus, identifying and understanding the parental factors that affect sources of information can contribute to the development of strategies for promoting the knowledge and acceptance of pediatric vaccination among parents. Fourth, parents’ motivation to have their child vaccinated against COVID-19 has not been examined using a validated scale. The Motors of COVID-19 Vaccination Acceptance Scale for Parents (P-MoVac-COVID19S) [46] is a new instrument developed for measuring parents’ motivation to have their child vaccinated against COVID-19 and appears to be a good candidate instrument to assess individuals’ motivation to be vaccinated against COVID-19.

Comment

  1. sections of Introduction. The background of the COVID-19 pandemic seems too simple. The background of the impact of the COVID-19 pandemic needs to be considered. There is a need to better elaborate the background of the COVID-19 pandemic. Please consider citing following papers: entitled "A preliminary assessment of the impact of COVID-19 on environment–A case study of China", and entitled "What does the China's economic recovery after COVID-19 pandemic mean for the economic growth and energy consumption of other countries?".

Response

Thank you for your suggestion. We added a new paragraph to describe the background of the COVID-19 pandemic as below. We also cited the two papers that you suggested into the revised manuscript (references 5 and 6). Please refer to line 50-59.

The coronavirus disease 2019 (COVID-19) pandemic has had a serious impact on human health and daily life. According to World Health Organization (WHO), by July 14, 2023, more than 760 million people have been confirmed to contract COVID-19; nearly 7 million people have died from COVID-19 infection [1]. Survivors of COVID-19 are at-risk for a variety of post-acute sequelae of COVID-19. A meta-analysis on 41 published studies demonstrated that the global estimated pooled prevalence of post-acute sequelae of COVID-19 was 43% [2]; fatigue was the most common symptom reported, followed by memory problems [3]. In addition, the COVID-19 pandemic has negatively impacted the accessibility and structure of education, availability and prices of food, economic activities, mental health, and quality of life in people around the world [46].

Comment

  1. section of Method. This section offers a detailed description. It is impressive. However, it would be better to highlight your improvement of the method and your innovation in methods.

Response

We added the description highlighting the innovation in methods as below. Please refer to line 116-122.

Fourth, parents’ motivation to have their child vaccinated against COVID-19 has not been examined using a validated scale. The Motors of COVID-19 Vaccination Acceptance Scale for Parents (P-MoVac-COVID19S) [46] is a new instrument developed for measuring parents’ motivation to have their child vaccinated against COVID-19 and appears to be a good candidate instrument to assess individuals’ motivation to be vaccinated against COVID-19.

Comment

  1. section of Result. This section is well-organized and well-written. It would be better to discuss what your findings are different from the past works. Deep and systematic analysis about the results should highlight the novelty of this manuscript.

Response

We added new contents describing the differences in the results between the present and previous studies as below.

The present study revealed that parents obtain information on pediatric COVID-19 vaccines from multiple sources. This result was congruent with the result of a previous in Chinese parents [33]. This study further extended the scope of the study to examine the various effects of information obtained from different sources on parents’ motivation to have their child vaccinated against COVID-19.” Please refer to line 244-248.

A previous study in Polish parents categorized television and the Internet as the same source of information [40], whereas the present study found that the influences of information obtained from the traditional and social media on parents’ motivation to have their child vaccinated against COVID-19 varied; moreover, the influences of information obtained from acquaintances and strangers on social media on parents’ motivation were also different.” Please refer to line 285-290.

Comment

  1. There are still some occasional grammar errors through the manuscript especially the article ''the'', ''a'' and ''an'' is missing in many places, please make a spellchecking in addition to these minor issues.

Response

We invited an English native editor to edit the manuscript. The grammar errors and typos were corrected. Attached please find the certificate for the edition.

Round 2

Reviewer 2 Report

The authors have incorporated comments from the first round of review. My concerns from my previous review have been addressed. I would recommend the paper to be accepted for publication.

The authors have incorporated comments from the first round of review. My concerns from my previous review have been addressed. I would recommend the paper to be accepted for publication.